# Changes in Gut Microbiota after a Four-Week Intervention with Vegan vs. Meat-Rich Diets in Healthy Participants: A Randomized Controlled Trial

**DOI:** 10.3390/microorganisms9040727

**Published:** 2021-03-31

**Authors:** Eva Kohnert, Clemens Kreutz, Nadine Binder, Luciana Hannibal, Gregor Gorkiewicz, Alexander Müller, Maximilian Andreas Storz, Roman Huber, Ann-Kathrin Lederer

**Affiliations:** 1Institute of Medical Biometry and Statistics, Medical Center, Faculty of Medicine, University of Freiburg, 79104 Freiburg, Germany; kohnert@imbi.uni-freiburg.de (E.K.); ckreutz@imbi.uni-freiburg.de (C.K.); 2Institute of Digitalization in Medicine, Medical Center, Faculty of Medicine, University of Freiburg, 79110 Freiburg, Germany; nadine.binder@uniklinik-freiburg.de; 3Laboratory of Clinical Biochemistry and Metabolism, Department of General Pediatrics, Adolescent Medicine and Neonatology, Medical Center, Faculty of Medicine, University of Freiburg, 79106 Freiburg, Germany; luciana.hannibal@uniklinik-freiburg.de; 4Diagnostic & Research Institute of Pathology, Medical University Graz, 8010 Graz, Austria; gregor.gorkiewicz@medunigraz.at; 5Center for Complementary Medicine, Institute for Infection Prevention and Hospital Epidemiology, Medical Center, Faculty of Medicine, University of Freiburg, 79106 Freiburg, Germany; alexander.mueller@uniklinik-freiburg.de (A.M.); maximilian.storz@uniklinik-freiburg.de (M.A.S.); roman.huber@uniklinik-freiburg.de (R.H.)

**Keywords:** nutrition, gut microbiome, diversity, plant-based diet, *Lachnospiraceae*, *Coprococcus*

## Abstract

An essential role of the gut microbiota in health and disease is strongly suggested by recent research. The composition of the gut microbiota is modified by multiple internal and external factors, such as diet. A vegan diet is known to show beneficial health effects, yet the role of the gut microbiota is unclear. Within a 4-week, monocentric, randomized, controlled trial with a parallel group design (vegan (VD) vs. meat-rich (MD)) with 53 healthy, omnivore, normal-weight participants (62% female, mean 31 years of age), fecal samples were collected at the beginning and at the end of the trial and were analyzed using 16S rRNA gene amplicon sequencing (Clinical Trial register: DRKS00011963). Alpha diversity as well as beta diversity did not differ significantly between MD and VD. Plotting of baseline and end samples emphasized a highly intra-individual microbial composition. Overall, the gut microbiota was not remarkably altered between VD and MD after the trial. *Coprococcus* was found to be increased in VD while being decreased in MD. *Roseburia* and *Faecalibacterium* were increased in MD while being decreased in VD. Importantly, changes in genera *Coprococcus, Roseburia* and *Faecalibacterium* should be subjected to intense investigation as markers for physical and mental health.

## 1. Introduction

The gut microbiota is the entirety of all gastrointestinal microorganisms (bacteria, viruses, protozoa and fungi) resident in the human gastrointestinal tract [1,2]. An essential role of the gut microbiota in health and disease is strongly suggested by recent research [3,4,5]. Gastrointestinal microorganisms are directly promoted or inhibited by nutrients selecting a diet-related microbiota composition, which is further indirectly affected by known factors such as metabolic processes and immunological modulation [2,6]. Several studies have shown associations between a disease-related composition of the microbiota (also described as dysbiosis) and a wide spectrum of chronic diseases, including cardiovascular and metabolic diseases such as type 2 diabetes as well as psychiatric disorders [7,8,9,10,11]. Recent research suggests that diet might be one of the potential key drivers of microbiota-associated diseases [12,13]. Some diets, such as a vegan diet, are known to show beneficial health effects—for example, improvements in cardiovascular risk factors and decreases in inflammatory markers—but it remains unclear whether these effects are gut microbiota-driven or not [14,15,16,17,18]. The microbial composition of long-term vegetarians and vegans was reported to be significantly different to omnivores, but the specificity of this difference appears to be unclear as the composition of the gut microbiota differs between the trials [9,19,20]. Furthermore, the comparability of trials evaluating the long-term effect of VD is very limited as the composition of the gut microbiota is not only affected by diet but also by origin, sex, age, and pre-existing illnesses, among other factors [1,2].

Recent publications dealing with short-term changes in the gut microbiota after becoming vegan are widely lacking. In a small cross-over trial comparing nine healthy subjects on a pre-cooked meal vegan diet (VD) or a pre-cooked meal meat-rich diet (MD), diet was able to modify the gut microbiota rapidly and distinctly [12]. The plasticity of the gut microbiota is highly emphasized by recent research, and the applied diet is crucial for respective effects as a strong association between food choice and gut microbiota composition was found [21,22,23,24]. The above-mentioned cross-over trial evaluating the effect a pre-cooked meal diet is, therefore, most likely not comparable to a free food choice diet. The effect of a short-term free of choice VD on the gut microbiota of healthy participants remains unclear. An understanding of the gut microbiota’s changes after becoming vegan is essential as it is the first step to elucidate whether the health benefits of VD might be attributed to a diet-driven gut microbiota composition. Aiming to close this research gap, a trial was planned to investigate the effect on gut microbiota when healthy omnivorous participants were randomized to a free of choice VD or to a free of choice MD for four weeks.

## 2. Materials and Methods

The monocentric, controlled, randomized trial with a parallel group design with healthy participants was conducted between April and June 2017 at the Center for Complementary Medicine, University Medical of Freiburg, Germany. The study protocol was approved by the ethical committee of the University Medical Center of Freiburg, Germany (EK Freiburg 38/17), and was performed according to the principles of the Declaration of Helsinki. The trial was prospectively registered with the German Clinical Trial register (DRKS00011963). All participants gave written informed consent before inclusion in the study.

Study population. Healthy, normal-weight, omnivorous subjects between 18 and 60 years of age, living in Freiburg for more than 6 months, without regular intake of medication, and with no clinically relevant allergies or food intolerance, were enrolled. Exclusion criteria were a history of eating disorders or being on a plant-based diet, participation in another clinical trial, and blood donation in the 4 weeks before the start of the trial, as well as the use of recreational drugs, nicotine, or alcohol. Participants had to be able to speak and understand German and to complete a nutritional protocol. Subjects were recruited via bulletins and newspaper announcements, and potentially eligible subjects were invited for a personal visit to check eligibility criteria in detail.

Intervention and control. For standardization, all subjects had to follow a one-week-long run-in phase with a balanced mixed (omnivorous) diet according to the recommendations of the German Nutrition Association (DGE) [25]. Afterwards, participants were randomly assigned to either a meat-rich (MD) (>150 g of meat per day) or a strict vegan diet (VD) for four weeks. The randomization list was created electronically block-wise (block size 13; Python Software) by a third independent person, and sealed envelopes were used for implementation. Every participant received extensive training on his/her assigned diet and detailed written information including a recipe book. No meals were provided, and participants were free to choose their food within their assigned diet. All of the participants had to fill out a weekly nutritional protocol, which was used to evaluate dietary adherence. Moreover, all of the participants had to keep their weight stable as weight changes are known to influence gut microbiota [26,27]. In case of weight loss, participants were recommended to incorporate high-caloric foods, whereas weight-gaining participants were advised to avoid high-caloric foods. Weekly follow-ups were scheduled between the study staff and the participants by phone or e-mail.

Fecal samples were collected in a special stool collector after the run-in phase but before being assigned to VD or MD (“baseline”) as well as after finishing the trial (“end”). Participants received the collectors after inclusion and were told to fill them immediately before meeting the study staff. Stool samples were frozen and stored at −80°C after receipt. Analysis of stool samples was performed at the Medical University Graz, Austria, according to an established protocol [28]. In brief, DNA of samples was extracted by mechanical lysis with a MagnaLyser Instrument (Roche Diagnostics, Mannheim, Germany), followed by subsequent total bacterial genomic DNA isolation with the MagNA Pure LC DNA Isolation Kit III (bacteria, fungi) in a MagNA Pure LC 2.0 Instrument (Roche Diagnostics) according to the manufacturer’s instructions. Template-specific sequences, targeting the hypervariable region V4 of the 16S rRNA gene, were used for amplification of bacterial 16S rRNA gene. Sequencing was performed with the MiSeq Reagent Kits v3 (600 cycles, Illumina, Eindhoven, Netherlands) according to the manufacturer’s instructions, with 20% OhiX (Illumina).

Fasting blood samples were taken after the run-in phase but before being assigned to VD or MD as well as after finishing the trial. Serum from each participant was aliquoted in 1.5-mL, non-diet-labeled cryovials. All methods were previously established and validated. Measurement of concentration of thrombocytes, neutrophilic granulocytes, and monocytes was performed by the Central Laboratory of the University Medical Center of Freiburg. Measurement of branched-chain amino acids was performed by the Laboratory of Clinical Biochemistry and Metabolism, Department of General Pediatrics, University Medical Center of Freiburg.

### 2.1. Bioinformatics

Sequences were processed within the QIIME2 framework [29]. The demultiplexed data were converted to the QIIME2 specific qza format and summarized with the demux summarize function. For quality control, the DADA2 plugin was used [30]. Within the denoise-paired function, sequences were truncated at position 290 due to a rapid drop-off of the quality score. Additionally, chimeric sequences were filtered out. Based on the DADA2 representative sequences, a phylogenetic tree was generated with the align-to-tree-mafft-fasttree function. Taxonomic assignments were obtained with the pre-trained Naïve Bayes classifier, trained on the Greengenes gg13_8 operational taxonomic units (OUT) reference database with 99% sequence similarity and the q2-feature-classifier plugin.

All downstream analyses were conducted in R-Studio (R v.3.6.3) [31]. Data were imported into a phyloseq object from the phyloseq R-package [32]. Results are presented at amplicon sequence variant (ASV) level. ASVs that were abundant in less than 5% of all samples were filtered out. Additionally, a genus level data set was created by agglomerating the data at genus level. For all statistical tests, a significance level of 5% was applied.

### 2.2. Data Analysis

#### 2.2.1. Alpha Diversity

To assess the alpha diversity of the bacterial communities, commonly used similarity indices were evaluated, considering both richness and evenness, to describe the diversity within samples. For species richness, the Chao1 index, a metric based on the number of observed taxa, was calculated. Taking microbial diversity and richness into account, the Shannon index was calculated, a metric based on the weighted abundance of microbes in each sample. The Inverse Simpson index, considering relative abundance, and Fisher’s index, quantifying the relationship between number and abundance of species, were also calculated. All indices were computed on the unfiltered ASV-level data. To test for differences in alpha diversity between baseline and end samples in each diet, a paired Wilcoxon signed rank test was applied.

#### 2.2.2. Beta Diversity

Principal coordinate analysis (PCoA) was used to visualize differences in community composition among the two diets and their respective baselines. To assess microbial similarity between samples based on the abundance profiles of shared taxa between each pair of samples, Bray–Curtis dissimilarity was applied. Moreover, weighted and unweighted UniFrac distances were calculated to quantify similarity based on (weighted) phylogenetic relationships between each pair of samples [33].

#### 2.2.3. Logistic Regression

To test for differences in the proportion of samples in which a taxon was detected, logistic regression was applied. The effects of diet, time, and the interaction of diet and time were estimated. The main effects in each diet assessed the change over time within a diet, e.g., “enriched in MD” or “depleted in MD”, while the interaction effects assessed the difference in one intervention compared to the other, e.g., “enriched in MD and depleted (or constant) in VD”. *p*-Values were corrected for multiple hypothesis testing with the Benjamini–Hochberg method and are referred to as p_adj_.

#### 2.2.4. ZINB

To identify differentially abundant bacterial taxa in both diets with respect to their baseline, a zero inflated negative binomial regression model (ZINB) was utilized. This model from the pscl R-package [32] takes the excessive zero inflation as well as the overdispersion in the data into account. In this two-part mixture, the model inflation of zero counts is modeled via logistic regression while the overdispersion in the count distribution is modeled as a negative binomial. In the log link function, the total sum of each sample is included as an offset, correcting for variation in library size. To test for diet-specific changes in microbial abundance, the effects of diet, time, and their interaction were estimated, enabling the detection of the main effects in each diet as well as the difference in one intervention compared to the other (interaction). *p*-Values were corrected for multiple hypothesis testing with the Benjamini–Hochberg method and referred to as p_adj_.

#### 2.2.5. PICRUSt

Metagenomic functions were predicted with the PICRUSt2 algorithm [34]. Based on the ASV abundances from 16S rRNA sequencing, KEGG orthologs, EC numbers, and MetaCyc pathways were inferred. Within the MaAsLin2 framework, a linear regression with default settings was applied to test whether specific metabolic pathways were enriched by VD or MD [35].

## 3. Results

Out of 150 interested persons, 53 were randomized and started the trial. The study flow is shown in Figure 1. Twenty-six participants were allocated to VD and 27 participants were allocated to MD for four weeks. All participants completed the study as per protocol.

Descriptive data of all participants are shown in Table 1. Baseline values did not differ significantly between the groups. Intake of energy and of fat measured by self-reported nutritional protocol during the trial did not differ significantly between VD and MD. Intake of carbohydrates (VD: 276.0 ± 85.1 g, MD: 241.5 ± 91.8 g, *p* = 0.001), protein (VD: 79.5 ± 28.5 g, MD: 112.4 ± 44.4 g, *p* < 0.001), and fiber (VD: 45.7 ± 19.5 g, MD: 24.9 ± 11.1 g, *p* < 0.001) differed significantly between VD and MD. Extensive results of nutritional intake were published previously and showed additional significant differences in vitamin and micronutrient intake (copper, zinc, phosphor, folate, calcium, sodium, vitamin B2, niacin, vitamin B6, vitamin B12, vitamin C, and vitamin E) [36]. Forty-five participants (85%) originated from Europe, whereas eight participants were born in other countries (Japan, China, Tunisia, Namibia, U.S.A., Mexico, India) but had lived in Germany for several years.

### 3.1. Microbial Sample Composition—Alpha Diversity

Alpha diversity analysis did not differ significantly between VD and MD (Figure 2). For the bacterial richness of the samples, quantified by Chao1, we found no differences between baseline and end samples, neither for VD nor MD (Figure 2a, *p*_VD_ = 0.770, *p*_MD_ = 0.629). The Shannon index, considering richness and evenness within the samples, did not reveal any significant difference in baseline and end, neither for VD nor MD (Figure 2b, *p*_VD_ = 0.921, *p*_MD_ = 1.000). Moreover, the Inverse Simpson index, based on the relative abundance of species as well as Fisher’s index, quantifying the relationship between number and abundance of species, did not change between baseline and end samples in either diet (InvSimpson: Figure 2c, *p*_VD_ = 0.921, *p*_MD_ = 0.861; Fisher’s index: Figure 2d, *p*_VD_ = 0.822, *p*_MD_ = 0.600). Hence, MD and VD had only a minor impact on the heterogeneity of microbial composition, emphasizing the intra-individual stability of participants.

### 3.2. Microbial Sample Composition—Beta Diversity

Bray–Curtis dissimilarity revealed no clustering of the samples according to dietary intake and timepoint. For most participants, baseline and end samples appeared to be close together on the PCoA plot, pointing towards a highly individualized and heterogeneous microbial composition (Figure 3a). The similarity of two samples from one individual was maintained during the study.

Results of UniFrac distances are shown in Section 3.6.

### 3.3. Logistic Regression and Differential Abundance (DA) between MD and VD

Logistic regression and analysis of differential abundance revealed a significant difference in *Coprococcus* between MD and VD. The proportion of samples in which *Coprococcus* was detected increased in VD from 42% at baseline to 81% at the end of the trial (*p*_adj_ = 0.047), while in MD, it decreased from 69% at baseline to 50% at the end of the trial (*p*_adj_ = 0.672, Figure 3b).

Zero inflated negative binomial model was applied to analyze the abundance of each ASV, with diet, timepoint, and their interaction as predictors (Appendix A). Main effects as well as interaction effects were estimated. While the main effects assess the change over time within a diet, e.g., “enriched in MD” or “depleted in MD”, in contrast, the interaction effects assess the difference in one intervention compared to the other, e.g., “enriched in MD and depleted (or constant) in VD”. In VD, ASVs of genera *Alistipes, Bacteroides, Blautia, Coprococcus, Dialister, Dorea, Faecalibacterium, Phascolarctobacterium*, and *Ruminococcus* are enriched at the end of the trial compared to baseline, while ASVs of genera *Akkermansia, Bacteroides, Bifidobacterium, Clostridium, Coprococcus, Faecalibacterium, Roseburia*, and *Ruminococcus* are depleted (Figure 4a). In MD, ASVs of genera *Alistipes, Bacteroides, Blautia, Clostridium, Faecalibacterium, Megamonas, Roseburia*, and *Ruminococcus* are enriched at the end of the trial compared to baseline, while ASVs of genera *Bacteroides, Bifidobacterium, Blautia, Dialister, Faecalibacterium, Gemminger, Phascolarctobacterium, Prevotella, Ruminococcus*, and *Sutterella* are depleted (Figure 4b).

As for most ASVs, the signal resulted from less than 10% of all samples; we filtered for the most abundant ASVs observed in at least 40% of samples. These are *Faecalibacterium* (unspecified species) and *Roseburia faecis*, which are depleted in VD, and *Blautia* (unspecified species) as well as *Faecalibacterium* (another unspecified species), which are enriched in MD (marked in red in meta data column for sample proportion).

Bacteria with significant interaction terms of diet and time changed significantly in MD over time compared to their respective changes in VD (Figure 4, red asterisk and Appendix A). The signatures of *Bacteroides, Clostridium, Faecalibacterium*, and *Roseburia* were enriched in MD and depleted in VD after the trial (Appendix A, top panel). Multiple ASVs of genera *Bacteroides, Blautia, Dialister, Faecalibacterium*, and *Ruminococcus*, however, were depleted in MD but enriched in VD after the trial (Appendix A, bottom panel).

Analysis of differential abundance at genus level revealed *Megamonas* to be more abundant in MD at the end of the trial and *Dorea* to be more abundant in VD at the end of the trial compared to their respective baseline. Both of them are, however, rare genera, as they were only detected in 7% of all samples (Appendix A).

### 3.4. Association of Bacterial Changes with Inflammatory Markers

Previous analyses of our research group showed a strong correlation of neutrophilic granulocytes, monocytes, and thrombocytes with serum concentration of branched-chain amino acids (valin, leucine, and isoleucine) in VD [17]. Therefore, we analyzed the associations of these parameters with the changes in the gut microbiota in VD and MD by applying ZINB with each clinical parameter separately as a predictor. Within the significant results (*p*_adj_ < 0.05) (Appendix A), we found in VD for all clinical markers strong associations with the rare genus *Odoribacter*, abundant in less than 10% of all samples (Figure 5a), and in MD with the rare genus *Clostridium* (Figure 5b). Interestingly, in VD, all branched-chain amino acids (VAL, ILE, LEU) were negatively associated with *Coprococcus* and *Dorea* and correlated positively with *Megamonas*. A negative association of branched-chain amino acids with *Dorea* was also found in MD (Figure 5b).

### 3.5. From Abundance to Function

We did not find any MetaCyc pathways, EC numbers, or KEGG orthologs to be significantly enriched in VD or MD at the end of the trial.

### 3.6. Unweighted UniFrac Distance

As discussed in Section 3.2, between-sample correlations based on Bray–Curtis distance did not reveal differences based on diet. Interestingly, when clustering the samples based on phylogeny by applying unweighted UniFrac distances, we observed a split of the samples into two groups, hereinafter called Phylo1 and Phylo2 (Figure 6a). Considering additionally the phylogenetic relationship with the abundance of species, by applying weighted UniFrac distances, the split was not as pronounced as before but still visible (Appendix A).

Analyzing the sample composition within Phylo1 and Phylo2, we found that while baseline samples from VD and MD were almost evenly distributed within each of the two groups, there was a slight, but non-significant, shift in sample composition after the trial (Figure 6b, *p* = 0.130). While, in Phylo1, end samples from VD were overrepresented, in Phylo2, the shift was towards the MD end samples.

#### 3.6.1. Logistic Regression and Differential Abundance between Phylo1 and Phylo2

To further understand the nature of Phylo1 and Phylo2, we looked at differences in microbial composition by applying logistic regression. We found that the taxa *Coprococcus* and *Parabacteroides* were significantly less abundant in Phylo2 compared to Phylo1 (Figure 6c). While the genus *Coprococcus* was present in 86% of all samples in Phylo1, it was only detected in 48% of all samples in Phylo2 (*p*_adj_ < 0.001). Parabacteroides was present in 63% of all samples in Phylo1 and only in 23% of all samples in Phylo2 (*p*_adj_ < 0.001).

Next, we identified differentially abundant ASVs between Phylo1 and Phylo2 by applying a zero inflated negative binomial model. Multiple ASVs are more abundant in Phylo2 compared to Phylo1 (Appendix A). The majority belonged to the genera *Bacteroides, Faecalibacterium prausnitzii,* and *Blautia*. Under the assumption that ASVs presented in a large proportion of the samples are the most relevant, we filtered for those significantly differentially abundant ASVs that were also detected in at least 40% of all samples. Here, we found two ASVs of genus *Bacteroides* and four ASVs of genus *Blautia* that were more abundant in Phylo2 compared to Phylo1 (Figure 6d).

To see whether the above-mentioned observations were mainly driven by single ASVs or if they were a general effect of the particular genus, we agglomerated the data at genus level (Appendix A). We found *Alistipes, Bifidobacterium,* and *Dialister* to be significantly more abundant in Phylo2 compared to Phylo1 (Figure 6e).

#### 3.6.2. Association of Clinical Markers/Inflammatory Markers with Phylo1 and Phylo2

The splitting of samples in Phylo 1 and Phylo 2 was not significantly associated with the age (*p* = 0.263) or BMI (*p* = 0.454) of participants, but with gender (*p* = 0.042). In Phylo 2, female samples were overrepresented (71%), while in Phylo 1, gender was equally distributed.

Changes in the concentration of monocytes (*p* = 0.842), neutrophilic granulocytes (*p* = 0.775), and thrombocytes (*p* = 0.142) were not significantly associated with the splitting into Phylo 1 and Phylo 2.

## 4. Discussion

The difference in the gut microbiota after short-term nutritional change to VD or MD was not fundamental as sample composition did not differ significantly in terms of alpha and beta diversity, but we found a few, mostly highly individual-dependent differences between VD and MD in single ASVs. Most of our participants showed a similar microbiota composition at the start and at the end of the trial, emphasizing the intra-individual stability of the gut microbiota towards diet change. The observed changes in ASVs were attributed to a few participants, predominantly in ASVs occurring in less than 40% of samples. The remaining genera (*Coprococcus, Roseburia, Blautia*) were mainly from the family of *Lachnospiraceae*, which is an interesting observation. *Lachnospiraceae* are a family of anaerobic, fermentative, and carbohydrate-metabolizing bacteria probably playing a role in metabolic and inflammatory diseases [37].

Similar to our results, recent research evaluating the effects of commonplace plant-based diets failed to show differences in alpha and beta diversity and distinct microbiota alteration by diet [19,22]. At first glance, the results of our trial appear to be unusual as some recent publications suggest a distinct alteration of gut microbiota by diet [6,12]. Going more deeply into the alteration of gut microbiota by diet to explain the observed intra-individual microbiota stability of our participants, it is known that microbiota alteration depends not only on the type of diet but much more on the individual [2,38]. Johnson et al. examined the daily fecal samples of 34 healthy participants for 17 consecutive days and found that the daily microbial response was highly personalized [21]. Walker et al. showed in a clinical trial with 14 overweight young men that the gut microbiota of obese participants clustered more by individual than by diet. Furthermore, some types of bacteria have a greater response to diet changes than others, making some individuals, being home to these bacteria, more responsive to diet changes than others [13,39,40]. The extent of gut microbiota alteration by diet depends on the initial microbial composition [13]. The composition of bacteria is, in turn, an individual fingerprint [41]. In our trial, the individual response to nutritional change is emphasized by the significant change in a variety of ASVs, which resulted from less than 10% of all samples. Whether this individual gut microbiota response leads to an individual diet-related health improvement remains unclear and must be clarified by future research.

Nevertheless, our results and the results of other research groups suggest that VD led to distinct changes in single ASVs, being supposed to be crucial for health and disease [12,42]. *Coprococcus*, which was enriched in VD and depleted in MD in our trial, is reported to play a supportive role in mental health and is found to be depleted in the case of depression and in children with autism spectrum disorder [43,44]. A recent meta-analysis concluded that meat consumption might be associated with a moderately higher risk of depression [45]. Other recent publications found no relation between plant-based diets and mental health [46,47]. *Coprococcus* is also reported to be depleted in neurodegenerative disorders such as Parkinson’s disease [48]. Interestingly, twenty years ago, VD was discussed to be beneficial in Parkinson’s disease, but research data are still lacking [49]. VD is rich in polysaccharides, probably being causative for the enrichment of *Coprococcus* as it is reported that a higher intake of polysaccharides leads to a higher abundance of *Coprococcus* [50].

The role of *Blautia* in health and disease is controversially discussed as some species appear to be healthy while others are harmful. A higher abundance of *Blautia* is reported to be associated with Type 2 Diabetes and Hashimoto’s Thyroiditis, but one species, *Blautia obeum*, previously known as *Ruminococcus obeum*, appeared to be a sign of gut microbiota recovery after *Vibrio cholerae* infection in children [10,51,52]. Other species of *Blautia* are reported to be associated with obesity and metabolic inflammation [53]. In early childhood, the abundance of *Blautia* increases after cessation of breastfeeding and transition to solid foods, becoming a stable lifetime gut habitant. *Blautia* is reported to be enriched in children with phenylketonuria being on a nearly vegan, plant-based, low-protein diet, emphasizing its plasticity to adjust to dietary intake [54]. Nevertheless, the pathogenic impact of microbiota change by diet is difficult to elucidate as most of the observed changes, such as *Blautia* alteration, are part of a generally beneficial habitant network. *Roseburia*, for example, which was one of the remaining distinctive ASVs in VD and MD in our trial, is discussed to be a “marker of health”, being depleted in a variety of diseases [55,56,57,58]. Interestingly, the abundance of *Roseburia* decreased in VD in our trial, which was not expected. Contrary to our results, David et al. reported a significantly higher abundance of *Roseburia* in VD after short-term VD compared to MD [12]. Plant-based diets are rich in fiber, which is assumed to increase the abundance of butyrate-producing *Roseburia* [15,59]. The fiber intake of our VD participants was significantly higher than the intake of MD participants, but *Roseburia* abundance was higher in MD. The reason for this could be the choice of food. *Roseburia* is reported to be enriched by the intake of wholegrain foods [60]. It is possible that MD participants chose more wholegrain foods than VD participants as nutritional protocols did not clearly differentiate between wholegrain and non-wholegrain foods. A strong association of microbial composition with the choice of food is reported [21,22,23]. Many nutritional clinical trials limit the choice of food. For example, David et al. gave precooked meals made of vegetables, rice, and lentils to their plant-based participants, whereas their animal-based diet consisted of cooked pork and beef, cured meats, and cheese [12]. These massive changes in nutritional behavior, avoiding food diversity, might be also responsible for the reported effects. We aimed to imitate a commonplace diet, which is why participants were free to choose by themselves what to eat within their assigned diet. To avoid further confounders such as smoking, alcohol, drugs, and long-term medication, the inclusion criteria of our trial were strictly chosen. Recruitment bias cannot be ruled out as it is conceivable that persons interested in diet are more likely to participate in nutritional trials. However, exclusion criteria defined that interested subjects were only eligible for study inclusion if they did not have a history of eating disorders and were not already on a planted-based diet. The diet adherence of our participants is assumed to be good due to the results of the nutritional protocols and the analysis of vitamin B12 showing significantly lower levels of holotranscobalamin (holo-TC) in VD compared to MD [36]. Holo-TC is a widely-used marker of vitamin B12 status and reflects the intake of vitamin B12, which is deficient in VD [36,61]. Another strength of our study is that body weight was the same in both interventional groups at baseline and remained unchanged at the end of the study. Body weight change is known to be a confounder in gut microbiota studies [26,27]. To be able to broadly map transient changes in the gut microbiota, it would be desirable for further research to collect more stool samples at different time points.

Interestingly, when clustering the samples based on phylogeny, we found a split of samples into two groups, which we named Phylo1 and Phylo2. The split was not clearly associated with any of the measured sociodemographic or clinical parameters and could not be explained by the differentiation of enterotypes, as postulated by Peer Bork [62]. Most of the participants in Phylo1 were assigned to VD, and participants who were Phylo1 at the beginning of the trial changed to Phylo2, if they were assigned to MD. This is an interesting observation, possibly related to dietary composition; however, the results were only tendencies as they did not reach the level of statistical significance. Further research on larger cohort of individuals may elucidate whether Phylo1 versus Phylo2 splitting is a consequence of the nutritional composition of VD and MD or due to other yet unrecognized factors. Contamination of the samples by different researchers can be ruled out as all samples were processed by the same person, observing technical standards.

Finally, we highlight the limitations of our study. We evaluated the microbiome at only two time points, after a one-week run-in phase with a uniform balanced mixed diet and after the four-week-long intervention. Thus, our study does not provide insights into the dynamics of the microbiome composition induced by the run-in phase, nor does it provide a temporal resolution after the intervention. Therefore, rapid transient changes are not resolved by our study design. Moreover, due to the limited sample size (53 patients divided into two groups), small effects may not be significant due to limited statistical power. Moreover, clinical trials with fecal samples are always error-prone as the collection of samples depends on the compliance of participants, and the time between bowel movement and further processing is crucial for the results [1]. All of our participants were told to fill the stool collector immediately before meeting with the study staff, but it is realistic that not all participants were able to do so, even if they did not communicate this.

## 5. Conclusions

The gut microbiota of healthy participants was overall not remarkably changed after transition to VD or MD for four weeks. Analyses of alpha and beta diversity showed high inter-individual gut microbiota variation among participants. Several diet-related ASV changes were observed after the trial, but most of them were only detectable in a few of the samples, emphasizing the highly individual response to nutritional change. Three genera, namely *Coprococcus, Roseburia,* and *Blautia,* mainly from the family of *Lachnospiraceae*, were found to be different between VD and MD after the 4-week trial. These results are interesting as these genera are often discussed to be markers for physical and mental health. The results indicate that VD and MD might have a potentially beneficial effect, including on potentially harmful bacteria, but to a highly individualized extent. The results emphasize the necessity for further research considering a participant’s individual gut microbiota response.

## Figures and Tables

**Figure 1 microorganisms-09-00727-f001:**
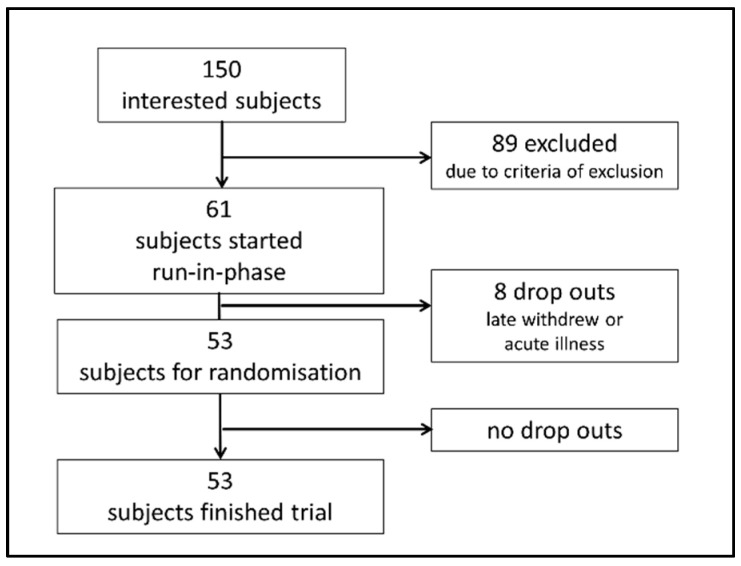
Flowchart visualizing recruitment of participants. Of 150 interested subjects, 61 were eligible for study inclusion and started a one-week run-in phase with a balanced mixed (omnivorous) diet according to the recommendations of the German Nutrition Association. Eight of these participants suffered from acute illness or withdrew from participation during the run-in phase and were not randomized. Fifty-three participants were randomized to VD or MD and started the trial. All randomized participants finished the trial as per protocol.

**Figure 2 microorganisms-09-00727-f002:**
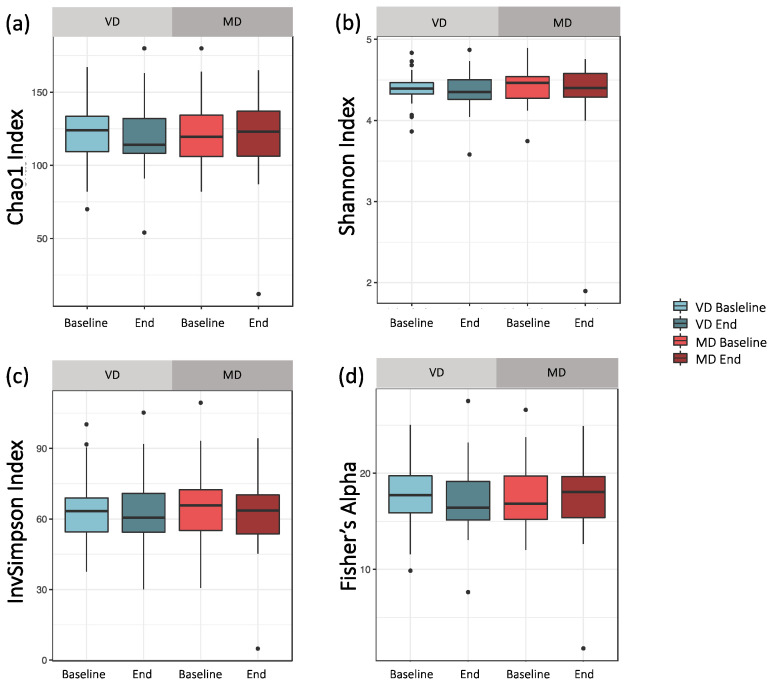
Comparing alpha diversity for VD and MD at baseline and end of intervention. (**a**) Chao1: Bacterial composition of samples based on the number of observed taxa. For both diets, the Chao1 index did not change after the trial (*p*_VD_ = 0.770, *p*_MD_ = 0.629). (**b**) Shannon index: By abundance weighted bacterial composition of samples, reflecting both richness and bacterial evenness within a sample. For both diets, the Shannon index did not change after the trial (*p*_VD_ = 0.921, *p*_MD_ = 1.000). (**c**) Inverse Simpson index: Species richness based on relative abundance. For both diets, the Inverse Simpson index did not change after the trial (*p*_VD_ = 0.921, *p*_MD_ = 0.861). (**d**) Fisher’s index: quantifying the relationship between number and abundance of species. For both diets, the Fisher’s index did not change after the trial (*p*_VD_ = 0.822, *p*_MD_ = 0.600).

**Figure 3 microorganisms-09-00727-f003:**
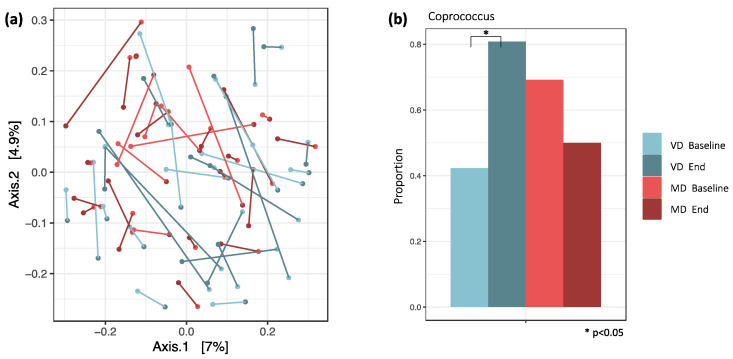
(**a**) PCoA plot with Bray–Curtis distances. Similarity of samples based on the weighted abundance of shared taxa. The connections between baseline and end samples indicate that both samples from one individual are very similar. (**b**) For VD and MD, the proportions of samples in which *Coprococcus* was detected are plotted. It was the only genus with a significant change in proportions between baseline and end samples in VD. Here, the proportion increased from 42% to 81% (*p*_adj_ = 0.047), while in MD, it decreased from 69% to 50% (*p*_adj_ = 0.672).

**Figure 4 microorganisms-09-00727-f004:**
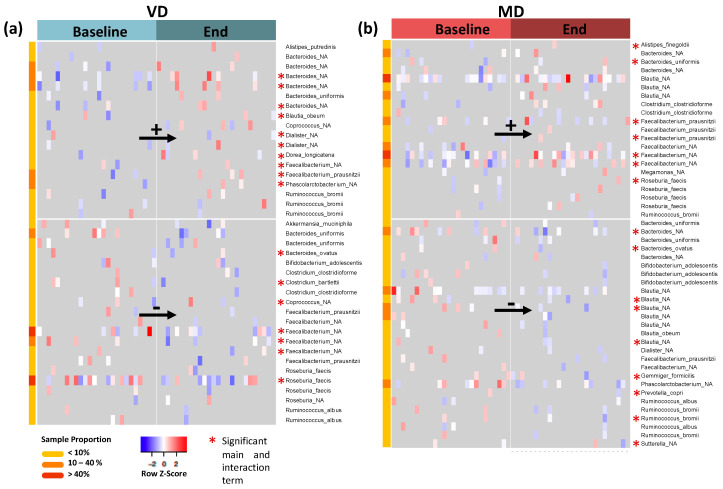
(**a**) Log-transformed and standardized abundances for all ASVs with significant main effects in VD; “_NA” describes unspecified species. Upregulated ASVs are shown in the top panel while downregulated ASVs are displayed in the bottom panel. The asterisk marks all ASVs which also have a significant interaction term, i.e., ASVs with a significantly different change in VD compared to their respective change in MD. (**b**) Corresponding plot for MD.

**Figure 5 microorganisms-09-00727-f005:**
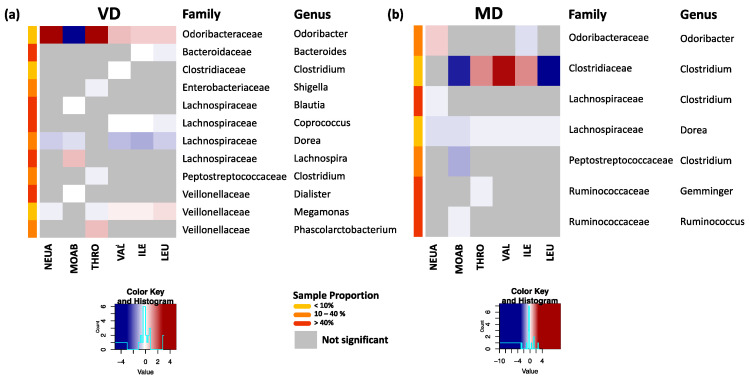
(**a**) Significant associations of clinical markers for neutrophils (NEUA), monocytes (MOAB), thrombocytes (THRO), and branched-chain amino acids valin (VAL), isoleucine (ILE), and leucine (LEU) with bacterial changes at genus level for VD. Here, the standardized estimates are plotted. Strongest associations with changes in the rare genus *Odoribacter* were observed. Highly abundant genera such as *Coporoccus*, *Dorea*, and *Megamonas* showed correlations with all branched-chain amino acids. (**b**) Corresponding plot for MD. All markers were associated with changes in *Dorea*, while the strongest associations occurred with the rare genus *Clostridiaceae–Clostridium*.

**Figure 6 microorganisms-09-00727-f006:**
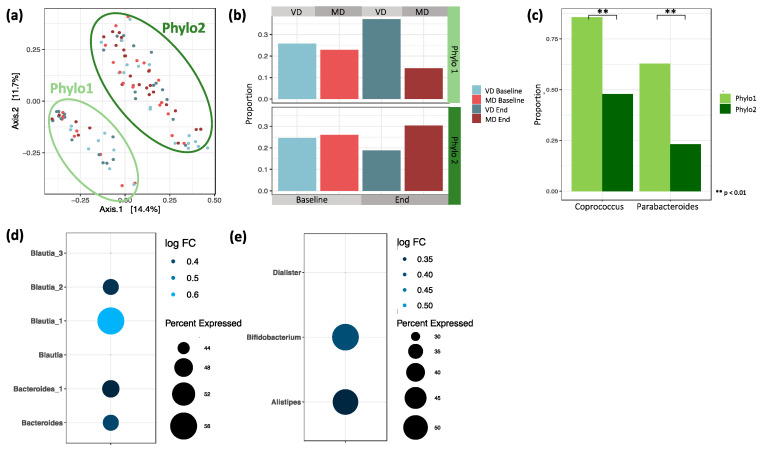
(**a**) PCoA plot based on pairwise unweighted UniFrac distances between samples. Similarity based on the length of shared phylogenetic branches between samples. (**b**) Proportion of baseline and end samples split by diet in Phylo1 and Phylo2. Phylo1 is enriched by end samples of VD and Phylo2 by MD (Chi-squared test, *p* = 0.130). (**c**) Logistic regression between Phylo1 and Phylo2 revealed two genera with significantly different proportions. *Coprococcus* differed significantly between Phylo 1 (86%) and Phylo 2 (48%, *p* = 0.006). *Parabacteroides* differed significantly between Phylo 1 (63%) and Phylo 2 (23%, *p* = 0.003). (**d**) ASVs detected in at least 40% of all samples that are differentially abundant between Phylo2 compared to Phylo1. (**e**) Genera differentially abundant between Phylo2 compared to Phylo1 after agglomerating ASVs at genus level.

**Table 1 microorganisms-09-00727-t001:** Demographic data of all participants in VD group and in MD group.

	VD (*n* = 26)	MD (*n* = 27)	*p*
Age ± SD (years)	33.2 ± 11.2	29.9 ± 9.5	0.407
Baseline: Body mass index ± SD (kg/m^2^)	22.9 ± 2.2	23.3 ± 2.6	0.444
End: Body mass index ± SD (kg/m^2^)	22.7 ± 2.0	23.4 ± 2.6	0.240
Sex (*n* male/*n* female)	8/18	12/15	0.229 *
Origin (*n* Europe/*n* other)	21/5	24/3	0.330 *

SD = Standard deviation, *p*-value depending on type of value and distribution from Mann–Whitney U-Test/Fisher’s exact test *.

## Data Availability

The data presented in this study are available on request from the corresponding author.

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
