# Peer review of "Changes in Gut Microbiota after a Four-Week Intervention with Vegan vs. Meat-Rich Diets in Healthy Participants: A Randomized Controlled Trial"

_microorganisms, 2021, doi:10.3390/microorganisms9040727_

Round 1

Reviewer 1 Report

Certain diets have a beneficial effect on health. However, the mechanism of their beneficial effects has not yet been fully elucidated. It remains to be speculated to what extent these effects can be attributed to the diet-dependent composition of the gut microflora. The presented research aims to search for and explain these dependencies. Despite the interesting content and methodology of the experiment, the manuscript has some inaccuracies that should be clarified and corrected.

Keywords: there are too many of them, the authors are asked to indicate a maximum of 6 most important references that characterize the subject of research. Keywords should be clearly formulated so that the reader can easily find the manuscript in the search engine after suspending the most important terms.

The authors are asked to specify the information contained in the diagram visualizing the recruitment of participants. In its present form, it raises some ambiguities.

Chapter: Conclusions. When analyzing the results contained in the manuscript, it should be stated that the summary does not cover all the most important achievements of the experiment. The authors are asked to formulate and indicate specific conclusions that result from the described research.

Author Response

Please see attachment, thank you!

Reviewer 2 Report

It is my pleasure to review this manuscript. The following are my comments and suggestion:

  1. Introduction is not well organized, some of contents could be moved to discussion.

Show the research gap for this study and clinical significance, a brief summary or overview of prior studies.

  1. Last sentence of introduction seems to indicate this trial as cross over trial. “change their diet to VD or MD”.
  2. The authors need to put the clinical trial registration information early in abstract and methods section for readers’ reference.
  3. For participants recruitment, will there be any potential selection bias, i.e study population might not represent the target population? This need to be discussed somewhere in discussion.
  4. I am not sure how would the trial keep all participants weight stable during the trial. It would be an intervention itself.
  5. I feel the 4 weeks intervention might be too short for gut microbiome change. Any power or sample size estimation for the trial?
  6. Several microbiome methods section is not very convincing, for example, Shannon diversity is more biodiversity, considering both richness and evenness; why only pick Shannon and Chao1, instead of other index, such as Inverse Simpson? Likewise, for beta diversity, why pick Bray-Curtis and unweighted UniFrac distance, instead of other indexes?  Why pick ZINB model, instead of using LEfSe analysis?  
  7. The whole microbiome analysis section needs more details and clarifications.
  8. Several results are not shown in any tables and figures, for example, the comparison of VD and MD. The authors need check it through in all manuscript.
  9. For “heterogeneity of microbial composition”, is it referred as “intra-individual variation’?
  10. Overall, the supplemental tables are not easy to follow.
  11. The figures are not easy to understand, especially on the interaction interpretation, what interaction is it? How did figures reveal the interaction?
  12. Overall, the results cover too much information, I would suggest the authors more focus on 2-3 major findings.
  13. Not sure on the Phylo1 and Phylo2, are there any clinical meaning, or just clustering by UniFrac Distance? One another way is using Dirichlet Multinomial Mixtures (DMM)(Quince et al. 2012) .
  14. The suggestion for results applies to discussion, too many findings make the discussion lose focus.
  15. For analysis on the taxa relative abundance, did the authors take consideration of multiple comparison/ multiple test issue?

Author Response

Please see the attachment, thank you!

Reviewer 3 Report

Kohnert et al determined the effects of vegan vs meat-rich diets on the gut microbiota in healthy individuals. Four weeks of dietary intervention had minimal effects on the overall gut microbiome structure and the authors attributed this to the intra-individual stability of the gut microbiota. As the authors noted in the introduction, there is a large body of literature investigating the effects of vegan diet on the gut microbiota. I am struggling to find the novelty and significance of the current study – the aim and hypothesis were not explicitly stated in the current manuscript. Limitations in the experimental design also appeared to significantly impact the quality of this work. My specific comments are as follows:

Major comments:

  1. To me a major pitfall of this study was insufficient timepoints in gut microbiota analysis. Did the authors collect fecal samples when the participants were on their habitual diet, i.e. prior to run-in? How did the diet, and therefore the gut microbiota, change when the participants transitioned from their habitual diet to the run-in diet? Although all participants were healthy individuals, one had to assume that the run-in diet would induce some changes in the gut microbiota, and therefore it was questionable to use the after run-in timepoint as the “baseline”. Also with only the before and after samples, the authors missed the opportunity to track changes in the gut microbiota during the 4-week vegan vs meat-rich dietary intervention.
  2. Please provide justifications for the sample size. While the authors were correct that intra-individual variations in gut microbiome could mask the dietary effects, sufficient sampling should still allow the study to detect diet-induced changes in the gut microbiota.
  3. The clustering of samples in the context of unweighted UniFrac distance was potentially interesting. The authors are encouraged to elaborate on that, particularly on potential mechanisms and physiological relevance. What about weighted UniFrac? Did you lose the clustering altogether?
  4. The discussion was somewhat “scattered” – the authors went into great lengths on a large number of individual bacteria but there was minimal interpretation of the overall findings and implications.

Minor comments:

  1. Please provide detailed dietary data during run-in and the interventions. Ideally we should also have the habitual diet data, i.e. prior to run-in.
  2. Was there any statistical analysis for beta diversity?

Author Response

Please see the attachment, thank you!

Round 2

Reviewer 2 Report

The authors have addressed all my comments and concerns well. 

Author Response

We are very happy about your response. Thank you again for your thorough review!

Reviewer 3 Report

Response and revisions from the authors are noted and greatly appreciated. While the clarity of the manuscript has largely improved, I continue to have significant concerns about the design of the study that negatively impacts the merits of the current work. Specifically, a key conclusion of this study was that the gut microbiome was “overall preserved” but this, as I noted in my previous review, could well be attributed to an insufficient sample size to account for confounders and therefore I have reservation on the scientific soundness of this finding. I do not necessarily agree with the authors’ notion that power calculation “was not feasible” because they did not have a single microbial species as the sole primary outcome or there was a lack of data on the effect size. It is of course obvious that power calculation for microbiome studies is not as straight forward as conventional clinical studies, however this is certainly doable, e.g. one could power for significant changes in metrices of overall microbiota structure. The authors should at least address this shortcoming in the manuscript and justify why they chose a sample size based on particular studies in the literature. Another key concern of mine is the design of the intervention – the duration and the expected dietary changes. As noted by the authors in the Introduction, the gut microbiota could be changed “rapidly and distinctly”. So what was the rationale for doing a 4-week intervention? With only the pre- and post-intervention samples, it was not possible to know how the microbiota responded to the changes – had it been fully transformed and remained stable after 4 weeks? Or it was still in the dynamic phase? Longitudinal sampling here is essential not simply because it allows us to characterize the dynamic changes, but to confirm that week 4 is the appropriate timepoint to assess the effect of the two diets on the gut microbiota. Finally, as noted in my previous review, data on the habitual and run-in diet are essential to characterize the extent of dietary changes from the habitual diet to the run-in and then experimental diet that contribute to changes in the gut microbiota. All the limitations above should at least be comprehensively addressed in the manuscript to be considered for publication.

Minor comments:

  1. It is confusing that section 3.2 was beta diversity but UniFrac distance, which is also a beta diversity metric, was in section 3.6.
  2. How often did the participants experience weight loss and increase their consumption of nuts or bacon? These were very specific but rather unusual food choices for weight management.

Author Response

Please see attachment, thank you!
